# PHARMACOPHORE-BASED DESIGN BY LEARNING ON VOXEL GRIDS

## ABSTRACT

Ligand-based drug discovery (LBDD) relies on making use of known binders to a protein target to find structurally diverse molecules similarly likely to bind. This process typically involves a brute force search of the known binder (query) against a molecular library using some metric of molecular similarity. One popular approach overlays the pharmacophore-shape profile of the known binder to 3D conformations enumerated for each of the library molecules, computes overlaps, and picks a set of diverse library molecules with high overlaps. While this virtual screening workflow has had considerable success in hit diversification, scaffold hopping, and patent busting, it scales poorly with library sizes and restricts candidate generation to existing library compounds. Leveraging recent advances in voxel-based generative modelling, we propose a pharmacophore-based generative model and workflows that address the scaling and fecundity issues of conventional pharmacophore-based virtual screening. We introduce *VoxCap*, a voxel captioning method for generating SMILES strings from voxelised molecular representations. We propose two workflows as practical use cases as well as benchmarks for pharmacophore-based generation: *de-novo* design, in which we aim to generate new molecules with high pharmacophore-shape similarities to query molecules, and fast search, which aims to combine generative design with a cheap 2D substructure similarity search for efficient hit identification. Our results show that VoxCap significantly outperforms previous methods in generating diverse *de-novo* hits. When combined with our fast search workflow, VoxCap reduces computational time by orders of magnitude while returning hits for all query molecules, enabling the search of large libraries that are intractable to search by brute force.

## 1 INTRODUCTION

Ligand-based drug discovery (LBDD) aims to identify and develop pharmacologically active compounds by making use of molecules that are known to interact with a given biological target (Kurogi and Guner, 2001). LBDD-based methods can be useful in the absence of the target's structure (Mason et al., 2001; González et al., 2009; Bernard et al., 2005; Loew et al., 1993), and can be complementary to structure-based drug design when target information is present, for tasks such as virtual screening and lead diversification (Grebner et al., 2020). Acharya et al. (2011) presents a detailed review of LBDD. Central to LBDD is the concept of a pharmacophore, which is defined as a feature of a molecule, located in 3D space, that may be involved in binding (Pearlman, 1993; McNaught et al., 1997; Chang and Swaan, 2006). Examples include hydrogen bond donors and acceptors.

Drug discovery campaigns typically rely on pharmacophore-shape based design for finding a diverse pool of molecules that are likely to bind to a target in the same way as a query molecule that has already been shown to bind. Having a diverse pool of binders increases the chances of finding a molecule that has minimal liabilities and has the potential to be optimised into a successful drug (Hawkins et al., 2007). This is conventionally done by running a brute force search against an existing library of molecules to find molecules whose 3D shape and arrangement of pharmacophore features in 3D space (i.e. pharmacophore-shape profiles) have high overlap in 3D space with the pharmacophore-shape profile of the query molecule (Rush et al., 2005); such molecules are known as hits with respect to the query molecule. The process involves three distinct steps: (i) generating conformations for the query molecule and all molecules in the library, (ii) computing the pharmacophore-shape profiles of these molecules, and (iii) aligning these profiles in 3D space and computing the overlap. This workflow (particularly the steps post-conformer generation/retrieval) is typically done using software such as

OpenEye's Rapid Overlay of Chemical Structures (ROCS) (Rush et al., 2005; OpenEye, Cadence Molecular Sciences, 2024).

There are two main issues with the prevailing pharmacophore-based virtual screening workflows. First, the process of aligning the query and library molecules and computing their overlap is time-consuming and expensive. This is particularly true because this process has to be carried out for each new query molecule in a project, of which we would ideally like there to be hundreds in the absence of major computational constraints. This bottleneck limits the number of query molecules that are screened in a typical drug discovery project, limiting the chances of finding a binding candidate molecule with promising properties and minimal liabilities. Second, candidate generation for any query is limited to the same set of compounds that is the existing library. The practical advantage of library screening is that molecules in these libraries are usually far more synthetically accessible (either already in-stock in commercial or corporate collections, or easily synthesizable via established routes and readily accessible building blocks) than *de-novo* generated molecules. However, this constraint does prevent the discovery of new compounds that may have better properties for the given task than those already present in the library.

In this work, we propose a generative model and workflows to address these issues. We make use of recent advances in voxel-based generative modelling, which have demonstrated the success of processing and generating molecules in voxel space (Pinheiro et al., 2023; 2024). Concretely, we propose *VoxCap*, a voxel-based generative model for pharmacophore/ligand-based drug discovery. VoxCap is a voxel captioning method that generates SMILES strings (Weininger, 1988) from an input voxelised representation of a molecule. We use VoxCap to generate SMILES strings corresponding to an input voxelised pharmacophore-shape profile.

We propose and evaluate two workflows: *de-novo* design, where we generate new molecules that are not in the training or test sets, with high pharmacophore-shape similarities to query molecules from the test set; and *fast search*, where we use *de-novo* design in conjunction with a fast 2D substructure similarity search to quickly and efficiently find hits in 3D space from a given library. Our evaluation shows that VoxCap outperforms previous work, by up to an order of magnitude, on the number of diverse *de-novo* hits generated given a query molecule. We also find that VoxCap, in conjunction with our *fast search* workflow, reduces the computational time/cost of inference by orders of magnitude while returning an appreciable number of hits for each query molecule.

## 2 RELATED WORK

### 2.1 IMAGE CAPTIONING

Deep learning has significantly automated the generation of captions from natural images. Vinyals et al. (2015) proposed the first end-to-end deep learning model to generate a caption from an input image. They use a convolutional neural network (CNN) (LeCun et al., 1998) to extract features from an input image, and a long short term memory network (LSTM) (Hochreiter and Schmidhuber, 1997) to decode this representation into a caption. Prior to this, Mao et al. (2014) and Kiros et al. (2014) proposed models that separately encoded images and text to rank or generate new captions. Xu et al. (2016) built on Vinyals et al. (2015)'s model by introducing an attention mechanism into the model to attend to different aspects of the input image. Continuing in the path of moving to attention-based architectures, Li et al. (2019) proposed a captioning model based on the Transformer (Vaswani et al., 2017). VoxCap modifies such approaches to operate on 3D inputs, thereby producing strings from voxel grids.

### 2.2 VOXEL-BASED MOLECULE GENERATION

While point-cloud based molecular generation remains widely used, recent work has focused on voxel-based molecular generative models. Ragoza et al. (2020) first proposed a voxel-based generative model for molecules using a variational autoencoder (VAE) (Kingma and Welling, 2022). Skalic et al. (2019) proposed a VAE-LSTM based model to generate SMILES strings given an input voxelised atomic representation of a molecule. Pinheiro et al. (2023) proposed a voxel-based generative model based on the neural empirical Bayes framework (Saremi and Hyvärinen, 2019), implemented using a UNet-based (Ronneberger et al., 2015) architecture. (Pinheiro et al., 2024) extended this framework to the conditional setting for binder generation, generating voxelised molecules conditioned on the voxelised representation of a protein pocket. Ragoza et al. (2022) also proposed a binding-site

conditioned voxel-based generative model using a conditional VAE-based architecture (Kingma and Welling, 2022; Sohn et al., 2015).

### 2.3 PHARMACOPHORE-BASED MOLECULAR DESIGN

Typical small molecule pharmacophore-based virtual screening workflows involve searching existing databases for molecules that match the pharmacophore and shape profiles of various lead compounds. Rush et al. (2005) proposed software to define, align and compute the overlap of pharmacophore-shape profiles, as well as a workflow to apply this to LBDD. Grebner et al. (2020) studied the relationship between database size and matching molecules found. Skalic et al. (2019) generated SMILES strings from voxelised atomic density input and conditioning pharmacophore information, and evaluated these against reference sets using distributional metrics such as uniqueness, Fréchet ChemNet Distance (Preuer et al., 2018) and distributional scaffold frequency similarity. Zhu et al. (2023) proposed a conditional VAE-based model to generate SMILES strings conditioned on stochastically generated pharmacophore graphs. They carry out evaluation by measuring a graph-matching score between query and generated molecules, and measuring properties such as docking score of generated molecules against the pocket from which the hit molecule was obtained.

## 3 METHOD

VoxCap generates SMILES strings from an input voxelised representation of a molecule. Our task here is to generate SMILES strings from a molecule's pharmacophore and shape profile. Our 3D voxelised molecular representation builds on previous works on voxelised molecule generation (Pinheiro et al., 2023; 2024). We represent a molecule with a Gaussian-like density centered around the location of each of the pharmacophores of the molecule, with each pharmacophore type having a separate channel. We use 6 pharmacophore types: hydrogen bond donor, hydrogen bond acceptor, cation, anion, aromatic ring and hydrophobe. We also use an additional shape channel, in which Gaussian densities are centered at the 3D positions of each of the atoms in the molecule, but in one channel and with all densities having the same radius. An example voxelised pharmacophore-shape profile is given in Figure 1. Section B of the Appendix gives further details on the voxelisation.

### 3.1 TRAINING OBJECTIVE

Given a molecule $m$ including its conformation, from a dataset $M = \{m_n\}_{n=1}^N$, where $N$ is the number of molecules in the dataset, we compute its tokenised SMILES string sequence $\mathbf{u} = (u^{(1)}, \cdots, u^{(T)})$, where $T$ is the length of the sequence and each element of $\mathbf{u}$ is an integer $r \in \mathbb{Z} : 1 \le r \le R$ representing a token from the vocabulary $\mathcal{V} = \{v_r\}_{r=1}^R$ of size $R$. Each vocabulary element $v_r$ is a sequence of characters; we use the vocabulary and tokeniser provided in the DeepChem package (Ramsundar et al., 2019). We also compute the molecule's voxelised pharmacophore-shape profile $\xi = f_\xi(m) \in \mathbb{R}^{C \times d \times d \times d}$, with number of channels $C = 7$ and each side of the voxel grid having dimensionality $d$, using a deterministic non-differentiable function $f_\xi$.

Our objective during training is to predict the ground truth SMILES string sequence $\mathbf{u}$ from the input voxelised representation $\xi$. The task is to maximise the likelihood of the data with respect to the parameters $\theta$ of our model. Our model is autoregressive and factorises the likelihood of each datapoint as follows:

$$p(\mathbf{u}|\xi;\theta) = \prod_{t=1}^T p(u^{(t+1)}|\mathbf{u}^{(1:t)}, \xi; \theta). \qquad (1)$$

The loss $\mathcal{L}$ for each datapoint is thus given by:

$$\mathcal{L}(\mathbf{u}|\xi;\theta) = -\log p(\mathbf{u}|\xi;\theta) = -\sum_{t=1}^T \log p(u^{(t+1)}|\mathbf{u}^{(1:t)}, \xi; \theta). \qquad (2)$$

As implied by this equation, the model is trained using teacher forcing. At each time step $t$ the model uses the softmax function to output a distribution $p(u^{(t+1)} = v_r|\mathbf{u}^{(1:t)}, \xi; \theta) = \frac{\exp(f(\mathbf{u}^{(1:t)}, \xi; \theta)_r)}{\sum_{j=1}^R \exp(f(\mathbf{u}^{(1:t)}, \xi; \theta)_j)}$ where $f(.;\theta) : (\mathbb{R}^* \times \mathbb{R}^{C \times d \times d \times d}) \to \mathbb{R}^R$ is a neural network with parameters $\theta$ mapping a sequence

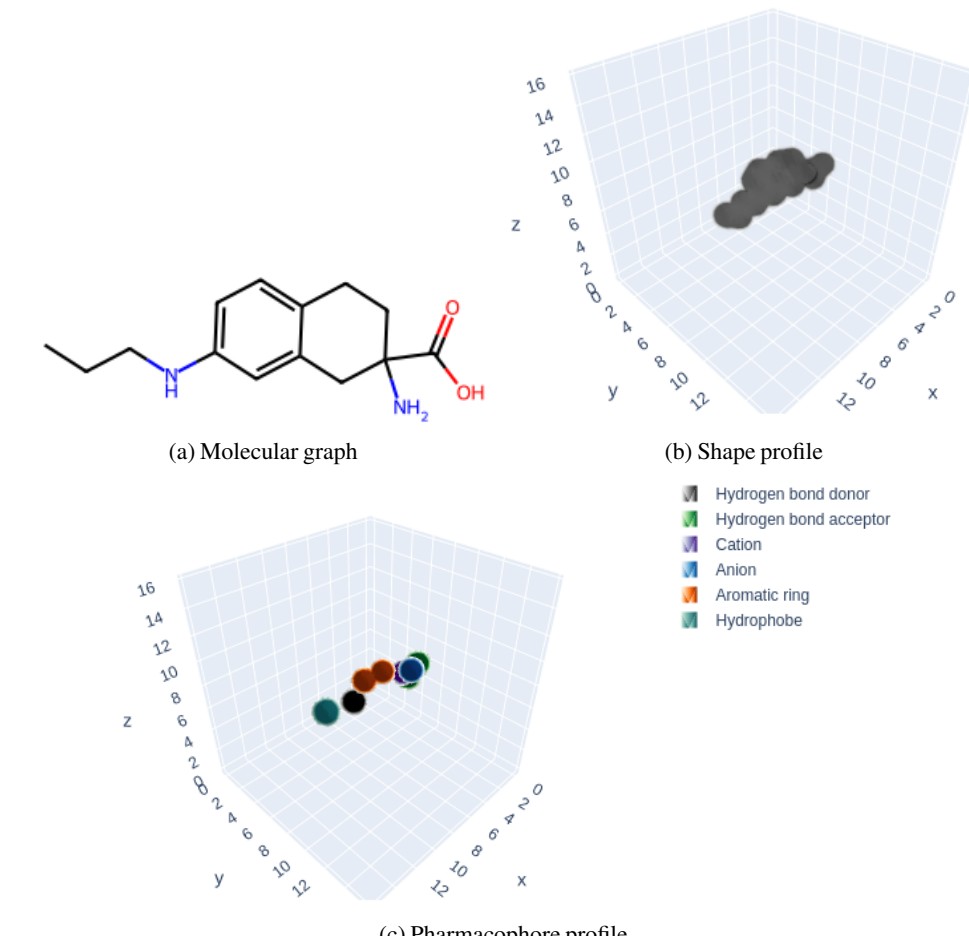

(a) Molecular graph        (b) Shape profile

☑ Hydrogen bond donor
☑ Hydrogen bond acceptor
☑ Cation
☑ Anion
☑ Aromatic ring
☑ Hydrophobe

(c) Pharmacophore profile

Figure 1: Example molecule from the ChEMBL dataset with its molecular graph, 3D shape profile and 3D pharmacophore profile. Each colour in the pharmacophore profile (one for each pharmacophore type as shown in the legend) is represented as a different channel to the model, as is the shape profile. Distances on the axes are measured in Angstroms. For visualisation purposes, we threshold the voxel grids displayed here such that values under 0.1 are set to 0.

of arbitrary length ($\mathbb{R}^*$ denotes a sequence of any length) and a voxel grid to a logit for each element in the vocabulary $\mathcal{V}$, and $f(.;\theta)_j$ is the logit output by this function for $v_j$.

## 3.2 GENERATION

At generation time, the voxelised pharmacophore-shape profile $\xi$ of a query molecule $m$ is passed as input to the model. We use ancestral sampling to generate SMILES strings $\mathbf{s}$ corresponding to new molecules, with a distribution $\tilde{p}$ over $\mathcal{V}$ that is annealed by a temperature parameter $\tau \in \mathbb{R} : \tau \geq 1$, as follows:

$$s^{(t+1)} \sim \tilde{p}(s^{(t+1)} = v_r | \mathbf{s}^{(1:t)}, \xi; \theta) = \frac{\exp(f(\mathbf{s}^{(1:t)}, \xi; \theta)_r / \tau)}{\sum_{j=1}^{R} \exp(f(\mathbf{s}^{(1:t)}, \xi; \theta)_j / \tau)} \tag{3}$$

## 3.3 MODEL ARCHITECTURE

We use an encoder-decoder model. As our encoder we use a 3D CNN-based architecture to compress the input voxels $\xi$, corresponding to a given molecule's pharmacophore-shape profile, into a vector representation. We then use this representation to initialise the hidden state of an LSTM decoder, and

also pass it in to this LSTM at every time step, concatenated with the embedded token for that timestep. A diagram of the model is given in Supplementary Figure 1.

## 4 EXPERIMENTS

### 4.1 DE-NOVO GENERATION FROM PHARMACOPHORE-SHAPE INPUT

We carry out experiments to evaluate the ability of our model to generate a diverse set of molecules matching an input pharmacophore-shape profile for the purpose of ligand-based drug discovery. To evaluate the overlap between the pharmacophore-shape profiles of two molecules, we compute OpenEye's ROCS Tanimoto Combo score. For the purposes of this paper, we abstract this as a function $TC(\cdot,\cdot): \mathcal{M} \times \mathcal{M} \to [0,2]$, where $\mathcal{M}$ is the space of all possible molecules with their conformations. TC hence takes in two molecules and outputs a score between 0 and 2, with 0 representing no overlap and 2 representing perfect overlap.

We carry out the evaluation on two datasets: *GEOM-drugs* (Axelrod and Gomez-Bombarelli, 2022) and the subset of *ChEMBL* (Mendez et al., 2019) defined in the GuacaMol benchmark (Brown et al., 2019), commonly used in the machine learning literature.

GEOM-drugs consists of approximately 430,000 drug-like molecules with up to 181 heavy atoms of type { C, O, N, F, S, Cl, Br, P, I and B }. The median number of heavy atoms per molecule is 45 and more than 99% of molecules have less than 80 heavy atoms. The dataset consists of multiple conformations for each molecule. We carry out experiments using different resolutions and grid sizes, and for our final model choose a grid of dimension $48^3$ with a resolution of 0.35 A. This volume covers over 99.8% of all points in the dataset. We perform the same preprocessing and dataset split as (Vignac et al., 2023), resulting in a 1.1M/146K/146K train/validation/test split.

The ChEMBL (subset) dataset consists of approximately 1,600,000 drug-like molecules with up to 88 heavy atoms of type { C, O, N, F, S, Cl, Br, P, I, B, Se and Si }. The median number of heavy atoms per molecule is 27. As the dataset is given as SMILES strings, we use RdKit (Landrum et al., 2023) to generate a 3D conformation for each molecule. We do this by using ETKDG (Riniker and Landrum, 2015) to get initial coordinates for the atoms followed by optimisation with reference to the Merck Molecular Force Field (Halgren, 1996). Our final model uses a grid of dimension $64^3$ with resolution 0.35. We use the same 1.3M/80k/240k train/validation/test split as Brown et al. (2019).

For each dataset, we carry out evaluation on 100 randomly selected 'query' molecules from the test set. A conformation is associated with each of these query molecules. As a baseline, for each of these query molecules we randomly sample 1000 molecules from the test set. We obtain multiple conformations for each of these sampled molecules using OpenEye, overlay them with the candidate molecule and compute the Tanimoto Combo score. For each sampled molecule, we retain the highest Tanimoto Combo score computed against the query amongst all conformations of the sampled molecule. We define a hit as a sampled molecule $m_{gen}$ with $TC(m_{gen}, m_{query}) \geq 1.2$ with respect to the query ligand $m_{query}$, in line with the heuristics used by Grebner et al. (2020). We evaluate on four metrics: number of hits per query molecule, number of hits per query molecule containing a unique scaffold, max Tanimoto Score per query molecule and number of query molecules with at least one hit.

We then evaluate our model as well as the Pharmacophore Guided Molecular Generation (PGMG) model proposed by Zhu et al. (2023) for pharmacophore-based generation. The PGMG model is based on a conditional variational autoencoder (Kingma and Welling, 2022; Sohn et al., 2015) architecture. At training time, the PGMG model learns to denoise a corrupted SMILES string conditioned on a pharmacophore point cloud that is computed from the uncorrupted SMILES string. At inference time, the model generates SMILES strings corresponding to a given pharmacophore point cloud. The point cloud is encoded using a Gated Graph Convolutional Neural Network (Bresson and Laurent, 2018) that is equivariant to rotation, and a transformer (Vaswani et al., 2017) architecture is used to process the encoded representation along with the SMILES string. For the ChEMBL dataset, we use the pretrained model provided by Zhu et al. (2023). For GEOM-drugs, we train a model using their codebase and use the best checkpoint output by their training script.

For each of the candidate molecules, we compute the pharmacophore-shape profile $\xi$ and pass it as input to VoxCap. We sample several SMILES strings from $p(\mathbf{s}|\xi;\theta,\tau)$ using different values of $\tau$ as well as top-k sampling (Fan et al., 2018). For the PGMG model, we pass the SMILES string of

| Method | Dataset | Hits | Unique Scaffold Hits | Max | # Queries with $\geq$ 1 hit |
|---|---|---|---|---|---|
| Dataset baseline | | 0 | 0 | 1.16 | 43 |
| PGMG | GEOM-drugs | 11.5 | 7 | 1.44 | 77 |
| VoxCap | | 116.5 | 55.5 | 1.77 | 97 |
| Dataset baseline | | 0 | 0 | 1.12 | 32 |
| PGMG | ChEMBL | 11.5 | 8.5 | 1.39 | 77 |
| VoxCap | | 115 | 72 | 1.77 | 95 |

Table 1: De-novo generation where the queries are drawn from a test set with the same distribution as the training set. Values for hits, unique scaffold hits and max score are medians across the 100 query molecules.

| Method | Training/Test Dataset | Hits | Unique Scaffold Hits | Max | # Queries with $\geq$ 1 hit |
|---|---|---|---|---|---|
| PGMG | ChEMBL/GEOM-drugs | 12 | 8 | 1.40 | 73 |
| VoxCap | | 85.5 | 48.5 | 1.65 | 90 |
| PGMG | GEOM-drugs/ChEMBL | 1 | 1 | 1.23 | 57 |
| VoxCap | | 23.5 | 12 | 1.47 | 77 |

Table 2: De-novo generation where the queries are drawn from a test set with a different distribution than the training set. Values for hits, unique scaffold hits and max score are medians across the 100 query molecules.

the query molecule into the model along with a pharmacophore graph computed from this string using the code provided by the authors, and sample several SMILES strings. Before evaluating, we remove invalid and duplicate generated molecules. We also remove generated molecules that are identical to the query molecule, as the purpose of this task is to generate novel, diverse molecules with a similar pharmacophore-shape profile to the reference molecule, not to reproduce the query molecule. For each model, we select the first 1000 valid, non-duplicate generated molecules corresponding to each candidate molecule, then obtain conformations and compute metrics for these molecules in the same way as for the dataset baseline. (For VoxCap, before selection, the generated molecules are arranged in ascending order of the $\tau$ value used to generate them, so that we favour selecting molecules sampled from the distribution on which the model was trained, where $\tau = 1$.) In this way, we compare approaches using the same budget for ROCS evaluation, which is the time-consuming part of the evaluation, compared to forward passes of neural networks which are relatively cheap.

We use the validation set for early stopping and model selection based on the total number of unique scaffold hits in a small set of 100 generated samples for each of 10 randomly selected validation set query molecules.

Results on GEOM-drugs and ChEMBL are given in Table 1. (See also Supplementary Table 1 for mean and standard deviation summary statistics.) Metrics corresponding to the dataset baseline are very low, showing that two randomly selected molecules from the same marginal distribution are expected to have low overlap. VoxCap outperforms PGMG by large margins across all metrics on both datasets. In contrast to PGMG, VoxCap produces at least one hit for almost every query molecule in each test set. Of note, along with the high number of hits, is the high number of unique scaffold hits, showing that the model produces a diverse set of hypotheses with matching pharmacophore-shape profiles, providing diverse avenues for ligand exploration corresponding to a given query. The top score among all generated ligands, averaged over all query molecules, is also higher for VoxCap than for PGMG across both datasets. VoxCap is therefore able to generate hits for queries drawn from the distribution on which it was trained. This in-distribution evaluation relates to drug-discovery campaigns where lead compounds are similar to library compounds that the campaign has access to, or has produced before, which have been used to train the model.

The mean and median values for all VoxCap metrics are comparable in all cases, showing that VoxCap's successful performance is not merely due to an extremely high contribution to the metrics from a small number of queries and low contribution from the rest, but is instead due to high contributions from many queries. The standard deviation across the 100 queries is, however, high for both the hits and unique scaffold hits metrics. Bringing this number down would correspond to more predictable performance for a given query and is an important area of future work.

We similarly measure out-of-distribution performance in Table 2 and Supplementary Table 2. For this task, models trained on GEOM-drugs are evaluated on query molecules from the ChEMBL subset, and vice-versa. We see that VoxCap once again outperforms PGMG across all metrics by large margins. VoxCap is hence able to generate hits for queries drawn from a distribution that is different to the one on which it was trained. This relates to drug-discovery campaigns where lead compounds are significantly different from compounds that the campaign has access to, or has produced before. Results from models trained on ChEMBL are better than the corresponding models trained on GEOM-drugs, possibly because of the larger number of molecules in ChEMBL.

## 4.2 FAST MOLECULAR LIBRARY SEARCH

Carrying out a brute-force ROCS search against a given database (also known as a molecular library) for each new query molecule in an experimental setting is a time-consuming and costly process. We propose an inexpensive approximation to this workflow by taking advantage of the generative capability and cheap inference cost of our model. Given a query ligand $m$ from the test set, we compute its pharmacophore-shape profile $\xi$, use VoxCap to generate several sample SMILES strings $\mathbf{s}$ from $p(\mathbf{s}|\xi;\theta,\tau)$, and then remove duplicate and invalid molecules. For each of these $n_g$ generated molecules we then find, in the union of the train and validation sets, the $n_a$ closest 2D analogs by Tanimoto similarity (i.e. Jaccard similarity) of Morgan fingerprints (a bit vector featurisation of a molecule, each element of which indicates whether a particular substructure is contained in the molecular graph), and compute the ROCS score for each of these analogs against the query molecule. A diagram of this algorithm is shown in Figure 2.

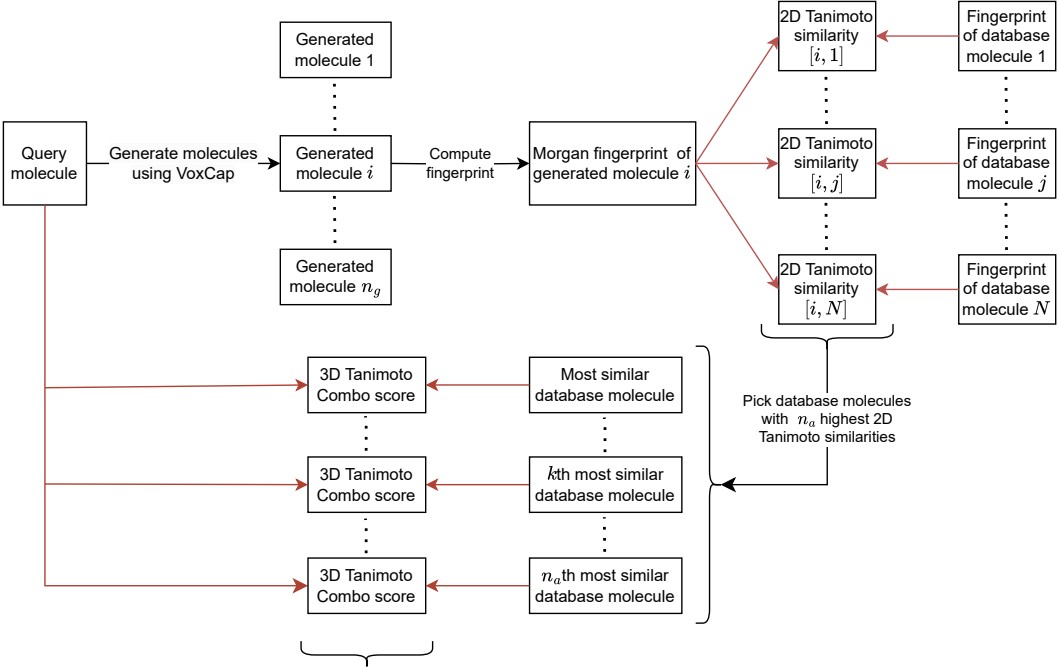

Figure 2: *Fast search* workflow, where we focus on the procedure applied to one of $n_g$ generated unique, valid molecules. Fingerprints of database molecules can be precomputed and reused for each new query to further reduce computational cost.

Our proposed approach drastically reduces the cost of database search. The 2D search we employ is fast and inexpensive, with negligible computational cost compared with 3D conformation generation, alignment to the query molecule and Tanimoto Combo score computation. Our approach brings down the number of 3D comparisons from $O(\text{database size})$ to $O(n_g \times n_a)$, a difference of several orders of magnitude depending on database size. As an example, the Enamine Real and Enamine Diverse databases consist of 60 billion compounds and 100 million compounds respectively, whereas $n_g \times n_a$ is typically on the order of 1000. Furthermore, running brute force 3D searches against the full Enamine

Real database is intractable, especially in a generative design framework where we wish to run searches for hundreds or thousands of generated molecules, whereas our fast search workflow is feasible. Our approach is advantageous to simply running the 2D search over the entire database for each query molecule as we are interested in finding compounds that are diverse in substructure space but similar to the query in 3D pharmacophore-shape space; not analogs that are simply similar to the query in substructure space.

We evaluate the accuracy of this workflow as follows. For each of 11 randomly selected query molecules from the ChEMBL test set, we run a ROCS search against every molecule in the union of the train and validation sets excluding the query molecule. We call this the brute force approach. We also use VoxCap to generate several molecules given the query molecule's pharmacophore-shape profile, and find either (i) $n_a = 5$ analogs for each of up to $n_g = 100$ unique, valid molecules, or (ii) $n_a = 1$ analog for each of up to $n_g = 500$ molecules. We compute these analogs' ROCs scores against the query molecule as described above. We then compute the number of hits in this set of analogs and compare with the number of hits in the set of top 500 molecules by ROCS score from the brute force approach for each query molecule. Results are given in Table 3.

We find that this workflow yields a significant number of hits and unique scaffold hits; the two settings of $n_g$ and $n_a$ yield similar results. The number of ROCS comparisons per query (up to a constant, $\alpha$, the average number of conformers per database molecule) is $n_g \times n_a$. This is several orders of magnitude below that of the brute force search, which is the size of the database. This difference would hence be even more pronounced in a real-world drug discovery application for databases such as Enamine Real/Diverse. We would also expect to find a greater number of 2D matches and hits from VoxCap-generated molecules for these databases due to their larger size. Our workflow hence enables the search of large-scale databases for sets of diverse hits, a current bottleneck in the drug design pipeline.

Our results show that there are limitations to our approach, namely the number of hits is over an order of magnitude lower than the brute force method, suggesting significant room for improvement. We explore why this may be the case. We observe that the highest 2D Tanimoto similarity for any given generated molecule is generally low, with a median top-1 similarity of 0.38 for the $n_g = 500, n_a = 1$ setting. Hence many of the generated molecules do not have close 2D analogs in the dataset. Supplementary Figure 2 shows a moderate correlation (Pearson $r = 0.4$) between 2D and 3D similarities. The lack of close 2D analogs for most of the generated molecules could therefore be a major reason for the low number of hits relative to the brute force approach. We also observe that there is a significant number of duplicate 2D hits per query, where the 2D search procedure maps on average 2.4 generated molecules to the same molecule in the database for the $n_g = 500, n_a = 1$ setting, thereby decreasing the number of hits that can be found on running ROCS.

We also observe from our experiments that our early stopping criterion selects a checkpoint after only a few epochs of training. Further training causes performance on sampling metrics to drop even as the validation cross-entropy loss continues to decrease. We hypothesise that the input representation is too specific to each individual molecule, so with continued training the model learns a distribution that is sharply peaked around the most likely molecule without adequately learning other modes of the distribution. This could mean that, at the early stopping checkpoint, when the number of training epochs completed is not large, the model has not been trained long enough to adequately capture the 2D structure of molecules in the dataset.

Future directions of work could involve regularisation/early stopping strategies that yield molecules more structurally similar to those in the dataset, as well as input representations that are smooth or noisy enough to facilitate the learning of several modes of the distribution over a larger number of training steps.

| Method | $n_g$ | $n_a$ | Hits | Unique | Max | Queries w/ $\geq 1$ hit | ROCS comparisons/query |
|---|---|---|---|---|---|---|---|
| Brute force | - | - | 450 | 151 | 1.67 | 11 | $\alpha \times 1.3 \times 10^6$ |
| VoxCap | 100 | 5 | 10 | 9 | 1.58 | 10 | $\alpha \times 500$ |
| VoxCap | 500 | 1 | 10 | 8 | 1.58 | 11 | $\alpha \times 500$ |

Table 3: Fast database search results. Unique stands for unique scaffold hits. ROCS comparisons/query are given including the constant $\alpha$, which is the average number of conformers per database molecule. Values for hits, unique scaffold hits and max score are medians across the 11 query molecules.

## 5 CONCLUSION

In this work, we propose the generation of molecules conditioned on voxelised 3D pharmacophore-shape profiles. We introduce VoxCap, a generative model that generates diverse SMILES strings from these pharmacophore-shape profiles. We propose a benchmark task for *de-novo* generation conditioned on pharmacophore-shape profiles and find that VoxCap outperforms the existing state of the art by large margins across all metrics. We also implement a generative model-based fast search workflow to practically enable the search of large libraries for structurally diverse 3D hits against a given query. We find that VoxCap, paired with this workflow, returns hits from the database for each query molecule, but that there is significant room for improvement. We identify areas of future work corresponding to coarser pharmacophore-shape representations and regularisation, that would enable the efficient search of large-scale databases for hit compounds in drug discovery campaigns.

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

# A    MODEL ARCHITECTURE

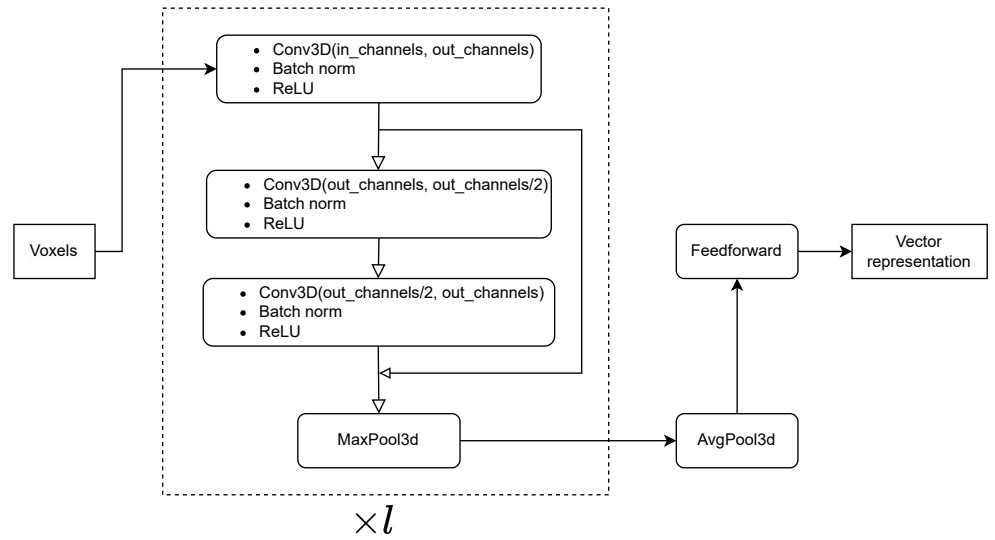

(a) The encoder compresses the input voxelised pharmacophore-shape profile into a vector representation. The 3D convolutional block is applied $l$ times, where $l$ is a hyperparameter.

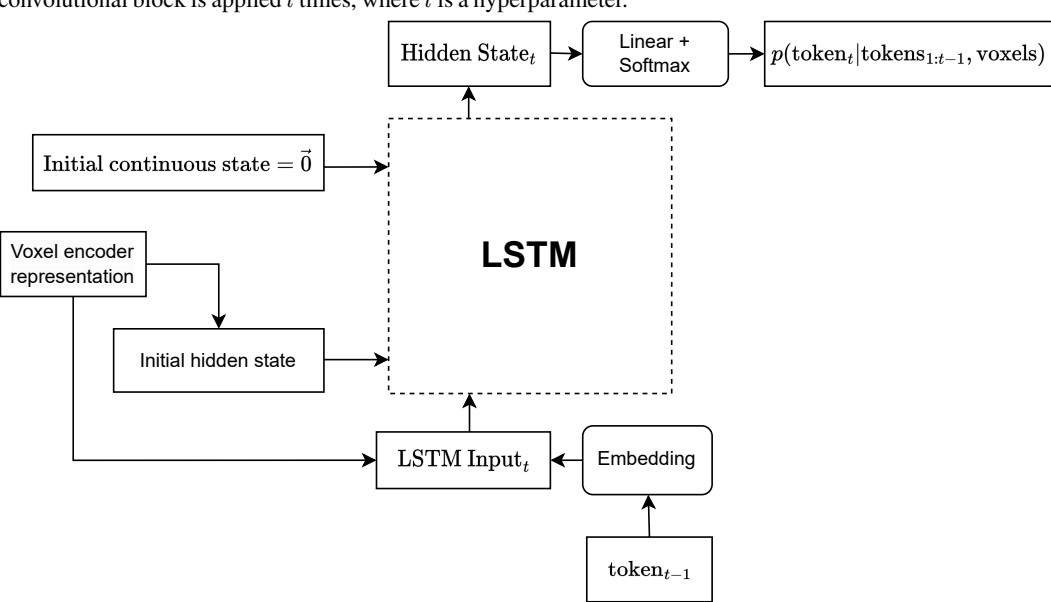

(b) The decoder maps the encoder's output to a tokenised SMILES string.

Supplementary Figure 1: Model architecture

## B  Voxelisation

We represent pharmacophores and molecular shape as voxelised densities. Given an input molecular conformation, we compute the locations of all pharmacophores in the molecule, resulting in a point cloud for each of the 6 pharmacophore types. We also compute a shape point cloud, whose points' locations are the same as those of the molecule's atoms. We now have a set of 7 point clouds $\mathcal{C} = \{\mathcal{C}_c\}_{c=1}^7$. The $c$th point cloud is given by $\mathcal{C}_c = \{a_{c,n}\}_{n=1}^{N_c}$ where $a_{c,n}$ is the $n$th point in point cloud $\mathcal{C}_c$ and $N_c$ is the number of points in point cloud $\mathcal{C}_c$.

We voxelise the pharmacophore and shape point clouds in a similar manner to how Pinheiro et al. (2023) voxelise atomic densities. First we convert each point $a_{c,n}$ into a 3D Gaussian-like density:

$$V_{a_{c,n}}(d,r) = \exp\left(-\frac{d^2}{(.93 \cdot r)^2}\right), \qquad (4)$$

where $V_{a_{c,n}}$ is defined as the fraction of occupied volume by point $a_{c,n}$ at distance $d$ from its center. We use radius $r = 1$ for all points. We then compute the occupancy of each voxel in the grid at each of the grid points $(i,j,k)$ at each channel $c$:

$$\text{Occ}_{c,i,j,k} = 1 - \prod_{n=1}^{N_c} \left(1 - V_{a_{c,n}}(\|C_{i,j,k} - x_{a_{c,n}}\|, r)\right), \qquad (5)$$

where $C_{i,j,k}$ are the coordinates $(i,j,k)$ in the grid and $x_{a_{c,n}}$ is the coordinates of point $a_{c,n}$. Since we use $c$ to index both channels and point clouds, points in point cloud $\mathcal{C}_c$ are assigned to channel $c$. Each voxel in the grid has a value between 0 and 1. We use the python package PyUUL (Orlando et al., 2022) to generate the voxel grids.

## C  Data augmentation

During training, we randomly augment each training sample. This augmentation comprises a random translation by a value uniformly sampled from [-1,1] for each axis, and a random rotation by a value uniformly sampled from $[0, 2\pi)$ for each of the three Euler angles. This augmentation is applied to the point clouds before voxelisation.

## D  De-novo generation

| Method | Dataset | Hits | Unique Scaffold Hits | Max |
|---|---|---|---|---|
| Dataset baseline | | $1.37 \pm 2.88$ | $1.33 \pm 2.75$ | $1.18 \pm 0.17$ |
| PGMG | GEOM-drugs | $20.14 \pm 23.76$ | $10.01 \pm 10.96$ | $1.41 \pm 0.30$ |
| VoxCap | | $126.08 \pm 98.22$ | $73.12 \pm 63.44$ | $1.69 \pm 0.24$ |
| Dataset baseline | | $0.63 \pm 1.25$ | $0.62 \pm 1.25$ | $1.12 \pm 0.15$ |
| PGMG | ChEMBL | $47.98 \pm 74.71$ | $27.56 \pm 37.83$ | $1.42 \pm 0.27$ |
| VoxCap | | $150.70 \pm 121.50$ | $90.69 \pm 80.00$ | $1.70 \pm 0.25$ |

Supplementary Table 1: De-novo generation where the queries are drawn from a test set with the same distribution as the training set. Values for hits, unique scaffold hits and max score are means $\pm$ standard deviations across the 100 query molecules.

| Method | Training/Test Dataset | Hits | Unique Scaffold Hits | Max |
|--------|----------------------|------|---------------------|-----|
| PGMG | ChEMBL/GEOM-drugs | $51.21 \pm 90.04$ | $27.73 \pm 43.86$ | $1.39 \pm 0.30$ |
| VoxCap | | $111.29 \pm 100.01$ | $67.56 \pm 65.54$ | $1.61 \pm 0.28$ |
| PGMG | GEOM-drugs/ChEMBL | $11.97 \pm 20.16$ | $5.97 \pm 9.93$ | $1.26 \pm 0.24$ |
| VoxCap | | $55.39 \pm 75.62$ | $33.63 \pm 49.02$ | $1.46 \pm 0.30$ |

Supplementary Table 2: De-novo generation where the queries are drawn from a test set with a different distribution than the training set. Values for hits, unique scaffold hits and max score are means $\pm$ standard deviations across the 100 query molecules.

## E    FAST DATABASE SEARCH

| Method | $n_g$ | $n_a$ | Hits | Unique Scaffold Hits | Max |
|--------|-------|-------|------|---------------------|-----|
| Brute force | - | - | $326.27 \pm 214.53$ | $158.82 \pm 121.47$ | $1.64 \pm 0.21$ |
| VoxCap | 100 | 5 | $20.64 \pm 17.55$ | $12.73 \pm 9.31$ | $1.52 \pm 0.55$ |
| VoxCap | 500 | 1 | $16.09 \pm 10.63$ | $11.82 \pm 7.68$ | $1.65 \pm 0.21$ |

Supplementary Table 3: Fast database search results. Values for hits, unique scaffold hits and max score are means $\pm$ standard deviations across the 11 query molecules

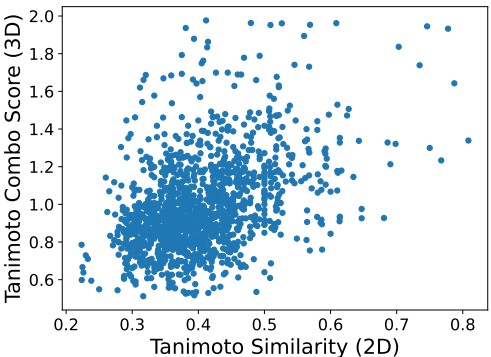

Supplementary Figure 2: 3D vs 2D similarities, from fast database search experiment with $n_g = 500, n_a = 1$

