# OpenReview forum: "Pharmacophore-based design by learning on voxel grids"
_ICLR.cc/2025/Conference — Submitted to ICLR 2025_

### Official Review · Reviewer_7k3F · 2024-10-23

**Soundness:** 2
**Presentation:** 3
**Contribution:** 1
**Rating:** 3
**Confidence:** 5

**Summary:**

The authors present VoxCap, a shape/pharmacophore-conditioned generative model for molecules that generates SMILES strings with an LSTM while conditioning on voxel representations of a reference molecule’s 3D shape and pharmacophores. The authors evaluate VoxCap in shape/pharmacophore-conditioned de novo molecular design tasks, principally evaluating against PGMG, which is a recent model that generates molecules conditioned on a graph-based representation of a reference set of pharmacophores. The authors show that VoxCap outperforms PGMG at generating molecules that are more similar to a reference ligand with respect to their shapes and pharmacophores, according to the external 3D scoring function ROCS. Conformers are enumerated for the generated SMILES prior to ROCS evaluation.

VoxCap also introduces a strategy of quickly searching an enumerated virtual chemical library for hits that are similar with regards to their shapes and pharmacophores compared to a reference molecule. Specifically, the authors first generate analogs of the reference molecule with VoxCap, and then use fast 2D chemical fingerprint similarities to search the virtual library for compounds that are graphically similar to the generated molecules. This strategy returns fewer hits with adequate ROCS scores compared to a ROCS-based virtual screen of the virtual library, but requires fewer ROCS evaluations.

**Strengths:**

**Originality and Significance**

The proposed VoxCap model empirically outperforms a recently introduced pharmacophore-conditioned generative model (PGMG) at generating new molecules that have adequately high 3D shape and pharmacophore similarities compared to a reference molecule, at least according to the authors’ presented evaluations.

**Quality**

Apart from some notable exclusions (see Weaknesses), the authors provide detailed explanations of the motivation, related work, and modeling task of VoxCap. The authors use a well-validated and popular cheminformatics tool (ROCS) for the majority of their evaluations of shape and pharmacophore similarity. The proposed machine learning modeling approach is reasonable.

**Clarity**

The paper is well written and easy to understand.

**Weaknesses:**

1. VoxCap appears to be a basic (re)implementation of previous published work on pharmacophore-conditioned SMILES-based generative models, without substantial modifications that are new or meaningful to the ICLR community. In particular, VoxCap looks quite similar (at least at a high-level) to LigDream (*Skalic et al., Shape-Based Generative Modeling for de Novo Drug Design, JCIM, 2019*), which used RNNs to caption voxel representations of shapes/pharmacophores. Note that LigDream was introduced 5 years ago. The authors should explicitly indicate how their proposed model meaningfully differs from existing work, and how VoxCap’s technical machine learning contributions contribute to its empirical or theoretical performance.

2. Evaluating PGMG with ROCS’ combined shape+color Tanimoto score is somewhat unfair, as PGMG does not explicitly condition on molecular shape (PGMG only considers pharmacophores, which only implicitly define the molecular shape via their spatial arrangement). It would be more appropriate for the authors to compare against other *existing* voxel-conditioned molecular generation approaches that *do* condition on both shape and pharmacophores (see further comments below).

3. Before the authors can claim SOTA in pharmacophore/shape-conditioned molecular generation, they need to compare against papers beyond PGMG, especially considering the rich literature on SMILES-based generative models that are explicitly designed to maximize pharmacophore and shape similarity. Many of these previous works are cited by the authors in their Related Work sections, but are not included as comparisons despite their high methodological relevancy. The original work in this area is LigDream (*Skalic et al., Shape-Based Generative Modeling for de Novo Drug Design, JCIM, 2019*), whose model design is quite analogous to VoxCap. I also highly recommend that the authors compare against *Papadopoulos et al., De novo design with deep generative models based on 3d similarity scoring, Bioorganic & Medicinal Chemistry, 44, 2021*, which uses REINVENT to generate molecules that directly optimize the combined ROCS color+shape score. REINVENT is widely used for general molecular optimization, and is often employed for pharmacophore and/or shape-conditioned molecular design tasks.

4. The author’s *fast search* approach --  which virtually screens an enumerated library for hits that are similar to a reference ligand in terms of their pharmacophores --  is not very well-motivated and empirically does not perform well.  The authors first generate *de novo* analogues of the reference ligand that are supposed to have high 3D pharmacophore similarity (but ideally low 2D chemical similarity) compared to the reference ligand, but then use (fast) 2D chemical similarity-based lookups to search the virtual library for molecules that are similar to the generated molecules. However, the entire point 3D pharmacophore-based virtual screening is to find hits that have disinct 2D chemical structures but similar 3D pharmacophores compared to a reference molecule, as 3D pharmacophore similarity and 2D chemical similarity may not be well correlated. The hits from VoxCap's virtual screen (which are identified via 2D chemical similarity to generated molecules) are not guaranteed to have high pharmacophore similarity to the original reference molecule, especially if the hits' chemical fingerprint similarities w.r.t. the *de novo* generated queries are low. Indeed, the authors report in Line 403 that these observed chemical similarities are relatively low. Hence, the proposed *fast search* approach is not likely to find many compounds that have high *3D pharmacophore similarity* compared to the original reference ligand. Indeed, this outcome is what the authors observe. Given 500 (generated) queries, only 10 hits (out of 450 total possible hits) were identified via 2D-nearest-neighbor-lookup that showed adequate 3D similarity to the reference ligand. It would be more principled to use a 3D pharmacophore/shape fingerprint and screen the virtual library based on *alignment-free* pharmacophore fingerprint similarity. This is indeed *many* previous works do to avoid the costs of alignment-based pharmacophore similarity searching -- these works are notably not cited by the authors; see below comment.

5. The authors ignore many previous works that have developed fast methods for pharmacophore-based virtual screening. A non-exhaustive selection of relevant works that develop pharmacophore fingerprints for this task include:
    - Pharmacoprint - https://github.com/lstruski/Pharmacoprint
    - PMapper - https://github.com/DrrDom/pmapper
    - Psearch -  https://github.com/meddwl/psearch
    - Pharmer - https://pubs.acs.org/doi/full/10.1021/ci200097m
    - RDKit’s native functionality: https://www.rdkit.org/docs/GettingStartedInPython.html#d-pharmacophore-fingerprints

6. The authors do not adequately evaluate the diversity of generated molecules, particularly the 2D chemical similarity of the generated molecules with respect to the original reference ligand. Although the authors do discard molecules that are exactly the same as the reference ligand in their evaluations, it is important to compare their Morgan chemical fingerprint similarities, as well, to ensure that the “hits” that are generated aren’t just slightly altered versions of the original molecule. Indeed, the maximum ROCS scores that the authors report from VoxCap are as high as 1.77, which in my experience only occurs when the two molecules under comparison are very similar in chemical graph space. The authors’ “unique scaffold hit” metric is inadequate for evaluating meaningful molecular diversity/novelty, as “unique” scaffolds can still be quite similar in chemical graph space. Evaluating global chemical fingerprint similarity would be more appropriate. Additionally, the authors should report validity, novelty, uniqueness, and diversity metrics to support this analysis.

7. No code or reproducibility statement is provided.

**Questions:**

- From a technical machine learning perspective *and* an empirical perspective, what does VoxCap contribute to the ML or drug design community that is not contributed by existing pharmacophore-conditioned SMILES-based generative models like LigDream?

- How many conformers are enumerated per generated molecule, for both VoxCap and PGMG, prior to ROCS-based similarity scoring? Are the conformational energies of these conformers considered?

- What is the correlation between (3D) ROCS Tanimoto Score and (2D) Morgan Fingerprint similarity between generated molecules and the reference ligand? Showing these distributions at a scatterplot would be helpful to evaluate the chemical diversity of generated compounds *versus* their ability to preserve the 3D pharmacophores/shape of the reference ligand.

---

### Official Review · Reviewer_auhf · 2024-10-28

**Soundness:** 2
**Presentation:** 2
**Contribution:** 2
**Rating:** 3
**Confidence:** 5

**Summary:**

This paper presents VoxCap, a voxel captioning method for translating voxel representations of 3D molecule pharmacophores to SMILES using a CNN-LSTM architecture. Experiments demonstrate usage for generative de novo design and fast search of molecular databases.

**Strengths:**

- This work builds upon 3D ligand-based virtual screening where the proprietary 3D shape matching software, ROCS, is commonly deployed and as such is highly relevant to practical drug discovery.
- The proposed method, VoxCap, generates molecules with substantial 3D shape and pharmacophoric similarity to query molecules surpassing the baseline PGMG in generating 'hits', 'unique scaffold hits' and maximising the shape similarity score to the query.

**Weaknesses:**

Major:

1. Experimental results could be strengthened:
  - De novo design
    - PGMG is a relevant baseline, however data processing differs between VoxCap and PGMG making it difficult to interpret which part of the workflow contributes to difference in performance. The authors could investigate a hybrid approach method which matches the conformer generation and voxelization approach. Or in general provide an ablation study that shows why VoxCap is better than other methods.
    - Comparison with other generative chemistry approaches such as REINVENT with a ROCS scoring function is necessary to interpret the fidelity of VoxCap.
    - In addition to the current dataset baseline, the authors could compare with the full database search (as in the virtual screening approach below), since this would demonstrate the shortcomings of virtual screening as mentioned in the introduction.

  - Fast search
    - The authors use VoxCap as a method for virtual screening but do not evaluate it using standard virtual screening benchmarks such as https://github.com/rdkit/benchmarking_platform.
    - To my understanding, the Brute force method serves as a source of truth in this experiment (an upper bound on performance), and therefore this experiment has no baselines. For example the authors mention morgan fingerprints, this could be added to Table 3 as a baseline.
    - Given the critiques surrounding the speed of ROCS, it would be sensible to include faster 3D shape comparison methods as baselines such as USRCAT and pharmit (or any other).

2. Claims surrounding performance
  - One of the main claims of the authors is that forward passes through a neural network are cheap compared with ROCS. This claim should be substantiated with numerical measures of performance. e.g. How long would it take to fast-search the molport or enamine REAL library.

Minor:
- The introduction focusses on the shortcomings of virtual screening in terms of molecule novelty, however as the authors point out (line 63) this has significant practical advantages and as such generative tools are not usually considered for this task as a matter of choice. The authors could reposition their argument such that both de novo design and virtual screening are both shown as useful parts of the drug discovery process.
- The authors definition of pharmacophore is inaccurate, there is a true pharmacophore (that may not be 3D and refers to the minimal features necessary for molecular recognition), and a computational pharmacophore model (features that have the potential for interacting, for which many models exist: CREDO, UFF, OpenEye, etc). This should be clarified.
- The authors should include rates of duplication and invalid SMILES removed before analysis.
- Some more insight into the properties of molecules generated by VoxCap would be appreciated, length distributions, molecule features, etc.
- Standard deviations in Table S1 should be included in table 1 and 2 since they are discussed in the main text.
- Consideration of prior work should be extended:
  - The authors focus on virtual screening, but VoxCap resembles a generative method and as such the paper would be improved by a discussion of generative modelling in chemistry, including common methods such as REINVENT (see https://www.sciencedirect.com/science/article/pii/S1359644621002531).
  - Amongst 3D generative models, discussion of molecular representation would be valuable since voxel based models have known advantages and disadvantages over e.g. graph-based 3D models (see https://www.sciencedirect.com/science/article/pii/S0959440X23000404).
  - The paper would also benefit from a discussion of how VoxCap differs from prior 3D ligand based pharmacophore models such as those presented in PGMG, Ragoza et al. and Pinheiro et al..
  - The speed of ROCS is a well known limitation of that tool, hence the development of many fast cheminformatic solutions such as FastROCS (GPU-accelerated ROCS), USRCAT (and other shape-based fingerprinting techniques) and pharmit (pharmer, https://pubs.acs.org/doi/10.1021/ci200097m). The latter two of which are available open source.

**Questions:**

1. Given that you have trained a model on the GuacaMol training dataset already, could you not perform the rediscovery and similarity tasks of the GuacaMol benchmark. This would give you results for an industry standard benchmark and a natural comparison to other Generative Chemistry models?

---

### Official Review · Reviewer_Dpeb · 2024-10-30

**Soundness:** 2
**Presentation:** 3
**Contribution:** 3
**Rating:** 5
**Confidence:** 4

**Summary:**

I appreciate the opportunity to review this manuscript on VoxCap, which presents a novel approach to address limitations in traditional pharmacophore-based virtual screening for drug discovery. The authors introduce VoxCap as a voxel captioning method for generating SMILES strings from voxelized molecular representations, presenting two workflows: de novo design for generating molecules with high pharmacophore-shape similarities, and fast search combining generative design with 2D substructure similarity search.

**Strengths:**

Novel combination of generative design and 2D substructure similarity search
Innovative voxel captioning approach for SMILES generation
Creative solution to address traditional pharmacophore screening limitations

**Weaknesses:**

Conceptual Framework and Definitions:
A concern lies in the manuscript's handling of core concepts. The authors' definition of a pharmacophore requires significant refinement. A pharmacophore isn't simply a "feature of a molecule" but rather a collection of essential features arranged in a specific 3D pattern. The current definition using "may be involved in binding" is too tentative - pharmacophore features are understood to be essential for binding. Furthermore, the statement that "Drug discovery campaigns typically rely on pharmacophore-shape based design" needs qualification, as many campaigns employ different approaches or pharmacophore representations beyond shape-based models. The manuscript needs to clearly distinguish between pharmacophore models and shape-based approaches, particularly when discussing their respective limitations.

Several technical aspects require clarification:
- The methodology for generating pharmacophore features isn't fully explained. The handling of heavy atoms versus hydrogens and pH/protonation states, which affect hydrogen bond donor/acceptor characteristics, needs elaboration. The manuscript would benefit from including distributions of pharmacophore features across the datasets in the supplementary information.
- Questions arise about feature assignment - particularly how aromatic and hydrophobic features spanning multiple atoms were handled. Was a Gaussian applied over a centroid within ring systems, or were features calculated individually for each atom? The consistent radius of 1 (presumably Angstrom) needs clarification, as does the handling of multiple pharmacophore features per atom.
- The treatment of SMILES representation isn't addressed. Whether canonical SMILES were used isn't specified, which is crucial given that molecules can have multiple valid SMILES representations. This choice could significantly impact training consistency.

A critical issue is the lack of code availability, which hampers reproducibility - a cornerstone of scientific research. The code should be considered an extension of the experimental setup, and its absence makes proper evaluation challenging.

The training methodology raises several concerns:
- The training objective focuses solely on SMILES string reconstruction without explicit optimization for pharmacophoric feature matching
There's potential redundancy in encoding pharmacophoric features when molecular shape often implies chemical group locations
The added computational complexity of incorporating pharmacophoric information may not justify the benefits
To validate the model's learning, I recommend testing three variants:
- The current implementation
- A version without pharmacophore features (shape-based only)
- A version with only pharmacophore features

Performance Claims and Evaluation:
The authors' claim of outperforming existing state-of-the-art "by large margins across all metrics" isn't fully supported by the data. The supplementary materials show high standard deviations, and no significance tests were performed. The "fast-search" approach, while novel, shows concerning underperformance in hit retrieval rates - a crucial metric in virtual screening.

Practical Applicability
The fast search workflow's practical utility is questionable. While generating similar compounds based on an "active" molecule is feasible, the current approach risks missing potential hits through its ECFP-based filtering. In drug development, where missing valuable analogues could cost months or years of development time, such limitations are significant. While computational efficiency is important, it shouldn't compromise thoroughness to this extent.

**Questions:**

1. How do you handle aromatic and hydrophobic features spanning multiple atoms?
2. Were canonical SMILES used in training?
3. Can you provide ablation studies comparing shape-only vs pharmacophore-only variants?
4. How do you justify the potential loss of valuable analogues in the fast search workflow?
5. Will you make the code available for reproducibility?
6. Can you provide statistical significance tests for the performance claims?

(See the rational above)

---

### Official Review · Reviewer_6n2D · 2024-11-04

**Soundness:** 2
**Presentation:** 2
**Contribution:** 1
**Rating:** 3
**Confidence:** 3

**Summary:**

This paper proposes a method named VoxCap for ligand-based drug design, which leverages recent advances in voxel-based generative modeling. VoxCap encode the molecular voxel with 3D CNN and generated SMILES with a LSTM decoder. The model can be used for de novo design, and equipping with a fast 2D substructure search algorithm, it can also be used in virtual screening on some ligand library.

The paper lacks novelty in methodology and the experiment is insufficient. I don’t think it meets the acceptance bar of ICLR. See below for detailed comments.

**Strengths:**

There isn't obvious strength. Weaknesses about the originality and significance are listed below.

**Weaknesses:**

* In the related work section, the authors pointed out some related work about voxel-based molecule generation. However, the authors didn’t discuss the difference between their proposed method and related work.
* From my opinion, the proposed method has very limited contribution, where  using 3D-CNN and LSTM as encoder and decoder to generate SMILES from molecular voxels has already well studied
* The figure 1 is unclear. Please use higher resolution figures and distinct visual markers for legends.
* It should be t=T-1 over sum symbol in Eq.1
* The paper only compare their method with a basic brute-force baseline and evaluate generated molecules from limited metrics. Current experiments are insufficient to demonstrate the proposed method’s contribution.

**Questions:**

(Have already stated in the weakness section.)

---

### Meta-Review · Area_Chair_kiqN · 2024-12-12

**Metareview:**

**(a) Summary**

This paper addresses the problem of ligand-based drug discovery. The core idea is to generate new molecules in SMILES representation while incorporating voxelized pharmacophore-shape profiles. The method generates SMILES sequences in an autoregressive manner by maximizing likelihood. The effectiveness of the proposed approach is evaluated empirically and compared to the PGMG method.

**(b) Strengths**

- **Technical contribution:** Combining 3D pharmacophore shape information with SMILES representation is a potentially effective approach.
- **Empirical performance:** The proposed method demonstrates improved performance over the well-established PGMG method in some metrics.

**(c) Weaknesses**

The reviewers have identified several significant weaknesses, which I summarize below along with my own opinions.
- **Motivation:** The paper lacks a strong motivation. Although voxel-based molecule generation methods exist, the specific advantages of the proposed approach over these existing methods are not well articulated.
- **Presentation and Quality:** The details of the proposed method are insufficiently described. Key aspects such as the formulation and training of the model remain unclear. While the authors mention using a 3D CNN-based architecture to embed 3D information and an LSTM for SMILES sequence generation, these descriptions are overly brief. This is particularly problematic as the process of integrating these components could represent a key technical contribution. Supplementary Figure 1 provides limited clarification.
- **Experimental design:** The experimental evaluation is incomplete, lacking comparisons with several recent methods. As a result, the reported performance improvements are not sufficiently convincing.

**(d) Reason of the decision**

The identified weaknesses represent significant limitations. In particular, the lack of clear methodological contributions is the most critical issue and alone justifies a recommendation for rejection.

**Additional Comments On Reviewer Discussion:**

No rebuttal was provided, and all reviewers recommended rejecting the paper without offering any further specific comments.

---

### Decision · Program_Chairs · 2025-01-22

Reject